# Consent for Teaching—The Experience of Pediatrics and Psychiatry

**DOI:** 10.3390/healthcare11091270

**Published:** 2023-04-28

**Authors:** Bárbara Frade Moreira, Cristina Costa Santos, Ivone Duarte

**Affiliations:** 1Faculty of Medicine, University of Porto, 4200-319 Porto, Portugal; 2Department of General and Family Medicine, Family Health Unit (USF) Caminhos do Cértoma, Grouping of Health Centers (ACES) Baixo Mondego, Regional Health Administration (ARS) Centro, 3050-428 Mealhada, Portugal; 3Department of Community Medicine, Information and Health Decision Sciences (MEDCIDS), Faculty of Medicine, University of Porto, 4200-319 Porto, Portugal; 4CINTESIS@REISE, Faculty of Medicine, University of Porto, 4200-319 Porto, Portugal

**Keywords:** informed consent, clinical teaching, medical students, medical education, medical ethics, pediatrics, psychiatry

## Abstract

Informed consent protects patients’ right of autonomy, as they may refuse to participate in clinical teaching. In Pediatrics, young people aged 16 or over, and with the necessary judgment, may consent; in Psychiatry, consent is also essential due to the personal nature of the subjects addressed. This study aimed mainly to assess the practical application of informed consent in medical education. An observational cross-sectional study was developed, and an interview-like questionnaire was applied to participants waiting for a scheduled consultation for themselves or the person they represented, in Pediatrics and Psychiatry. Only 54% of Pediatrics participants and 75% of Psychiatry participants stated that the physician asked them if they minded the students’ presence and an even smaller percentage from both departments affirmed that students introduced themselves as medical students and requested their consent to examine them. Patients feel satisfied to contribute to the students’ training, although a considerable percentage of them had experiences without being informed or asked for consent, which represents an evident disrespect for their autonomy. There is a need to intervene and provide an adequate education of ethical values in clinical practice to students.

## 1. Introduction

The importance of patients in medical education is widely recognized. Working with patients optimizes the clinical teaching of several skills, such as communication, observation and examination, which are essential for the future medical practice of the students [1].

The social, ethical and interpersonal dimensions of patient care are crucial to the acquisition of clinical skills and many medical schools have added programs that focus on Humanities, as well as inherent values and issues relevant to the profession [2,3]. The involvement of students in patient care is an integral part of medical education, as they are taught how to interview and examine real patients [4], learning, therefore, all of the three dimensions needed to manage a doctor-patient relationship: knowledge, skills and attitudes [5].

On the other hand, the involvement of medical students in the assessment and care of patients in teaching hospitals raises concerns that have ethical and possibly legal implications for the students as well as clinical teachers and teaching hospitals [6] [7]. Evidently, an ethical dilemma results from the fact that patients may not benefit from medical students participating in their care and can even be harmed by it [1]. However, this remains a relatively unresearched area [8].

The practice of medicine has drastically changed over the last years, as paternalism and the passive patient have slowly gone away [9]. Instead, the consent process has emerged, making the patient an increasingly more aware, fully informed and active participant in their health care provision [9]. This transformation in clinical practice patterns is aimed at respecting the Beauchamp and Childress’ Principles of Biomedical Ethics, which include the ethical principles of patient’s autonomy, beneficence and non-maleficence, and justice for all patients [10].

The principle of autonomy, as the right to self-determination, holds that individuals have the right to make their own choices, which speaking in terms of health care settings translates into the principle of informed consent [10]. Therefore, informed consent enshrines both principles of beneficence and respect for autonomy [6], since one of its main purposes is to protect the patient’s right of decision-making [11,12].

Just as with any medical procedure, clinical teachers and students are required to get a meaningful informed consent from patients or their legal representatives, so they can be involved and participate in the clinical teaching. In order to be an informed or valid consent, there must be a disclosure of an adequate amount of information, an understanding of that information and the voluntary choice of a preferred course [5,13]. This means that patients should understand what they are granting permission for, which implies a conversation, listening to patients, answering their questions and clarifying all their possible doubts [14].

In fact, interactions between patients and medical students can occur in a very busy environment, with clinical professionals under pressure, quick turnover of patients and, overall, scarce time and limited opportunities to ask them for consent [14]. However, there is a crucial requirement to respect patients and their rights [14], since their involvement should be guided by the principles of respect for individuals [1]. Therefore, it is important that patients know that they have the right to decide whether to participate in any medical educational activities, including the training of medical students [15], and that they are entitled to change their choice about the involvement of students and refuse it, at any time, without any negative consequences [14,16].

As patients may be reluctant or withdraw their consent at any time [17], gaining and maintaining the consent of a patient is a continuing process of communication and building trust [14]. For this reason, seeking for patients’ consent is also important for the practice of clinical skills, enabling students to establish patient rapport, and to allow patients tto vet the student, before consenting.

Regarding Pediatrics, medical students can acquire clinical skills by consulting and performing the physical examination, with children and their parents [18]. As Pediatrics naturally deals with underage patients, whose volitional skills are not yet fully developed, the power of decision-making in health matters rests with their legal representatives or guardians, who exercise the proxy consent. However, although there might be particular issues in obtaining consent from minors [7], the child’s perspective must be considered, in accordance with the concept of assent, which implies the guarantee of the minor [19]. Therefore, as the opinion of the minor is being taken into consideration as an increasingly determining factor [20], a shared decision should always be sought, integrating the parental permission and the child’s assent. Moreover, minors aged 16 or over, and with the necessary judgment to assess the meaning and scope of the diagnostic or therapeutic act proposed to them, may consent or dissent, regardless of their social, cultural and literacy characteristics. Accordingly, the Convention on Human Rights and Biomedicine states that the opinion of the minors should be considered, depending and in proportion to their age and degree of maturity, and that people with lack of ability to give informed consent should participate in the decision-making process, as far as possible [20].

In Psychiatry, the same can be verified in case of patients without the capacity to exercise the right of autonomy. According to the law, when it refers to adults that cannot consent to an intervention due to a mental disability, it may only be carried out with the authorization of their legal representatives or, if not possible, an authority or suitable person, acting on behalf of patient’s welfare [20]. Furthermore, regarding Psychiatry, the informed consent in medical education is also fundamental due to the level of sensitivity and personal nature of the subjects addressed and discussed in this medical care. These are factors that can, therefore, influence the acceptance of medical students’ involvement by patients and legal representatives or guardians in their health care.

This study aims to evaluate the practical application of informed consent and how patients or their legal representatives feel about the participation, involvement and conduct of medical students in their health care. As secondary objectives, this study also aims to compare these experiences between departments, by gender, age and level of education. Thus, this study intends to assess whether patients or their legal representatives exercise the right to autonomy, through the request of informed consent by medical students, in clinical teaching. In addition to the analysis of the current situation on this theme, these results, in two specialties with few studies, can alert decision-makers to the possible need of an intervention in the reformulation and rebuilding of these values in medical education, in order to promote the simultaneous improvement of the quality of the patient care and medical training.

## 2. Materials and Methods

This study took place over three months, at the University Hospital Center of São João, in the city of Porto, Portugal. This University Hospital Center is the main center in the north area of Portugal and one of the largest in this country, constituting a highly specialized and reference unit, and has the mission of providing the best health care to its community, promoting pre and postgraduate training and research, and always respecting the principle of humanization [21].

An observational cross-sectional study was developed, and an interview-like questionnaire was applied, in the form of a structured interview, to participants waiting for a scheduled consultation for themselves or the person they represented, in two distinct departments: Pediatrics and Psychiatry. This participation was voluntary; participants had time to reflect upon the request to participate and they could withdraw their participation and consent at any time. It was also explained that participants’ refusal or withdrawal would not harm their relationship with the investigator or the hospital. Moreover, the data was collected through an anonymous questionnaire, without any information that allowed the lead investigator to identify the participant, ensuring confidentiality. This study was approved by the Ethics Committee and Administration Board of the University Hospital Centre of São João and Faculty of Medicine of the University of Porto, on 26 November 2019.

Concerning Pediatrics, questionnaires were administered to patients’ legal representatives, except for those with ages between 16 and 18 years-old, for whom shared consent was obtained. In respect of Psychiatry, only adult patients were included. In the case of patients unable to answer the questionnaire, it was administered to their legal representative, while these patients’ assent was also obtained, as much as possible. All participants were interviewed by the same investigator, with 185 participants recruited, consecutively, per department, and each interview lasting, approximately, 10 min. Thereby, a total of 370 participants were interviewed, after being properly informed about the study’s goals and contents and obtaining their consent.

Participants were asked about their demographic information, such as gender and age, and then they answered specific questions concerning the involvement, or not, of medical students during their past appointments in this health care institution, followed by several closed questions, of yes or no, on their testimonies regarding this experience. These questions were based on several issues present in the literature and, evidently, participants only answered to the totality of the questionnaire if they had previous experiences involving medical students; otherwise, participants just provided their demographic information.

In case of legal representatives replying to the questionnaire, they only provided the patient’s gender and age, besides their own demographic information, and answered questions according to their own experience and that of the person they represented regarding medical students in consultations. Furthermore, since this questionnaire evokes past experiences, there was the option of “I don’t remember” in questions involving recourse to memory.

A data base was created with the obtained answers and statistical analysis was performed. Therefore, Statistical Package for Social Sciences (SPSS) was used to analyze the data and a *p*-value < 0.05 was considered significant. Additionally, Chi-Square Tests were used to compare some answers across patients’ age and education levels, according to the two different hospital departments. Independent sample T-Tests were also used to compare the mean ages between patients who answered affirmatively and negatively to some of the questionnaire’s questions.

## 3. Results

### 3.1. Description of the Participants

The sample consisted of 370 participants who were interviewed, namely 185 from Pediatrics and 185 from Psychiatry. Of these 370, 250 participants reported having participated in a previous consultation with the presence of medical students.

In Pediatrics, only 29 patients (16%) answered for themselves, while 156 participants (84%) were legal representatives (Table 1). Furthermore, 22 patients of these 29 and 108 legal representatives of the 156 had been in a consultation with medical students. There were 10 male and 12 female amongst the 22 patients; 11 patients were 16 years-old, 6 were 17 years-old and 5 were 18 years-old. Regarding the 108 legal representatives, 16 were male and 92 were female.

Concerning Psychiatry, 168 patients (91%) answered for themselves and only 17 (9%) respondents were legal representatives (Table 1). Moreover, 112 patients of the 168 and 8 legal representatives of the 17 had been in a consultation with medical students. Of these 112 patients, 23 were male and 89 were female; most were married (43%), while 37% were single, 2% widowed, 15% divorced and 3% in civil union.

### 3.2. The Experience of Pediatrics and Psychiatry with Medical Students in Health Care

Of the 185 interviewed participants in Pediatrics, 130 (70%) reported prior experiences with medical students in consultations, for themselves or the person they represented, mentioning a median of 2 students per consultation, with a minimum of 1 and a maximum of 6. Furthermore, of these 130, 22 (17%) answered for themselves and 108 (83%) were answered by a legal representative.

Concerning the 22 patients of Pediatrics who answered for themselves, only 11 (50%) reported that the physician asked them if they minded the students’ presence, 9 (41%) stated that students introduced themselves as medical students and requested their consent before performing the medical examination and only 7 (32%) reported that students explained to them the procedures they wanted to perform and addressed their doubts. The survey showed that 5 of them (23%) experienced difficulty disclosing an intimate problem of theirs in the presence of medical students and 4 (18%) would be more comfortable without the presence of students, if they could choose (Table 2). None of the participants claimed any situation of students’ disrespect towards them and all reported to feel satisfied to contribute to the training of medical students. In addition, no significant gender differences in the questionnaire responses were found among the patients who answered for themselves, nor was there a significative association with the gender of the children, in case of legal representatives responding to the questionnaire.

Regarding Psychiatry, of the 185 interviewed participants, 120 (65%) stated that they had participated in a consultation, for themselves or the person they represented, with medical students, indicating a median of 2 students per consultation, with a minimum of 1 and a maximum of 6. Moreover, of these 120, 112 (93%) answered for themselves and 8 (7%) responses were from a legal representative.

In Psychiatry, patients’ acceptance of medical students’ involvement in their medical care was found to be significantly associated with the patients’ age and education level. Actually, patients who felt uncomfortable with the students’ presence were younger (mean of 35 years old), than those who did not (mean of 49 years old) (*p* < 0.001) (Table 3). Additionally, a greater percentage of higher educated patients reported feeling uncomfortable with the students’ presence (34%) than patients who attended 1 to 4 years of school (0%) and those with 5 to 12 years of school (19%) (*p* = 0.003). Higher educated patients stated that they would be more bothered by the presence of medical students if their problem was related to an intimate part of their body (54%) than patients who attended 1 to 4 years of school (16%) and those with 5 to 12 years of school (31%) (*p* = 0.013) (Table 4).

Significant gender differences were also found regarding patients’ acceptance of students, among those that answered for themselves, in Psychiatry. Concerning the level of discomfort, a higher proportion of male patients claimed to feel uncomfortable with the medical students’ presence than female patients (*p* = 0.006). In fact, 14 male patients (61%) did not feel uncomfortable at all, but 9 of them (39%) felt moderately uncomfortable, although none felt very uncomfortable. 76 of the female patients (85%) did not feel uncomfortable at all, but 9 female (10%) felt moderately uncomfortable and 4 of them (5%) even felt very uncomfortable. Secondly, and concerning the nuisance that patients would feel if their problem was related to an intimate part of their body (*p* = 0.003), a greater percentage of male patients, namely 61% (14 patients), stated that they would be more bothered by the presence of medical students than female patients, with 28% (25 patients). Lastly, and regarding the choice that patients would make (*p* = 0.030), a greater percentage of male patients stated that they would be more comfortable without the presence of medical students, specifically 48% (11 patients), than female patients, with 25% (22 patients).

Besides, a significant difference between students’ conduct was found regarding patients’ education level, in Psychiatry. Students introduced themselves as medical students and requested patients’ consent to perform the physical examination to more than half of the patients who completed 4 or less years of school (52%) and slightly less to those with 5 to 12 years of school attendance (35%); however, this was reported in less than a quarter of patients with higher education (12%) (*p* < 0.001) (Table 4).

Table 5 compares participants’ experience with medical students in consultations between the two departments, since the department where participants were interviewed was significantly associated with some of their responses. Considering the 120 Psychiatry participants who had been in a consultation with medical students, 88 patients (75%), excluding the ones that didn’t remember, reported that the physician asked them if they minded that the students were present. 37 patients (31%), again excluding the ones that didn’t remember, stated that students introduced themselves as medical students and requested their consent to perform the medical examination; all 37 patients (31%) would be more comfortable without their presence. Participants did not report any situation of disrespect and all of them felt satisfied to contribute to the training of medical students, as well (Table 5).

Actually, Pediatrics participants reported a smaller percentage of certain practices, namely, the doctor asking the patient or the legal representative if the students could be present (*p* < 0.001 and *p* = 0.032, respectively) and introducing the medical students with their name and year of training (*p* < 0.001). On the other hand, more participants from Psychiatry stated that they or the person they represented felt uncomfortable with the students’ presence (*p* < 0.001 and *p* = 0.015, respectively). Psychiatry participants also reported a lower rate of medical students explaining to them the procedures they wanted to perform and addressing their doubts (*p* < 0.001), while a higher proportion of Pediatrics participants claimed to get more explanations about their illness or that of the person they represented, when students were present (*p* = 0.016). A higher proportion of Psychiatry participants would also be more bothered by the presence of medical students if their problem or that of the person they represented was related to an intimate part of their body (*p* = 0.012), although more participants from Pediatrics would be comfortable expressing this discomfort (*p* = 0.038). Lastly, more participants from Psychiatry claimed to have difficulty disclosing an intimate problem of theirs or the person they represented in the presence of medical students (*p* = 0.021) and reported that they would be more comfortable without their presence (*p* < 0.001).

## 4. Discussion

The experience of Pediatrics and Psychiatry regarding medical students’ involvement in patients’ health care was found to differ in several aspects. Firstly, only half of the Pediatrics department patients and three quarters of the Psychiatry department patients reported that the physician asked them if they minded the students’ presence. An even smaller percentage from both departments stated that students introduced themselves as medical students and requested their consent before performing the physical examination. These results are in accordance with previous studies, in which a significant number of patients claimed having past experiences with medical students, without being informed and asked for consent [16]. Therefore, although patients are a crucial part of clinical teaching, they might become less receptive to the medical students’ involvement due to the disrespect towards patients’ right of autonomy, by both students and their teachers.

Uncertainty can exist concerning who should introduce the medical student and inform the patients [16]. Actually, clinical teachers should consider identifying the student before the request of patient’s consent, but, according to The Council on Ethical and Judicial Affairs of the American Medical Association, students themselves should take responsibility for seeking the informed consent [17]. Although written consent may be medicolegally advisable, several studies found that patients did not consider it necessary [17], and considered that verbal consent is usually sufficient and adequate, in what concerns their participation in medical education [7].

It is always important to seek informed consent with courtesy and compassion for patients, taking into account their circumstances and vulnerabilities at the time [14]. However, the patient’s consent to have a student present during the consultation is, many times, sought at the last moment, making it quite difficult for the patient to decline, in the student’s presence [4]. Thus, patients should be previously informed, for instance, when the appointment is made [15], and the physician might even ask them privately, if they consent to the students’ presence [4].

As in this study’s findings, the literature shows that it is very common for patients to be misinformed [1]. In fact, informed consent requires that patients be explicitly told that medical students will be involved in their care and that they know the students’ level of knowledge and skill [6]; depriving them of having such data is not ethically accepted [5]. For this reason, patients should clearly understand what a medical student is [4] and recognize that students are not qualified doctors [22]. Thereby, patients must be informed about the students’ level of training, experience [11] and, ultimately, which procedures they might perform, before voluntarily consenting to their involvement [5].

There might be many explanations for the lack of disclosure towards the interviewed patients. To begin with, students may sometimes assume that the disclosure requirement has already been satisfied or even fail to recognize that they have an ethical obligation to obtain informed consent for their involvement in patient care [11]. However, even if there was no risk of harm from the students’ involvement in medical care, patients have the right to know who is providing their care [5]. On the other hand, several institutions may rely on a blanket consent, once students are considered part of the health care team and presumed consent covers their involvement in patient care. Still, it is inappropriate to assume that all patients would consent the students’ participation, just because they were admitted to an academic medical center [5]; hence, blanket consent should be discarded in favor of more adequate practices [6]. In addition, some students and teachers may believe that obtaining the patients’ consent would remove numerous educational opportunities, as too many patients would not allow medical students to take part in their medical care [5]. However, there is no empirical evidence in the literature that supports this claim [11].

In Pediatrics, interviewed participants were the ones reporting a smaller incidence of informed consent practices, in particular, by the physician. These participants were also the ones who claimed to get more explanations, in the presence of medical students, which is in accordance with previous studies, as patients reported finding out more about their condition when a student was present [15]. Furthermore, Pediatrics participants would express more their eventual discomfort, which may be related to the legal representatives’ desire to protect their children from any kind of their suffering.

Secondly, judgement and experience are needed regarding children under 16 years old. In fact, although most of the parents and children are prepared to be examined by medical students and become involved in medical teaching, it is always important that accurate and timely information is given to parents and children, so that there is a genuine opportunity for consent or refusal [18]. According to the literature, one possible explanation for why many children under the age of 16 are often not asked for consent is related to the fact that some physicians and students may believe that it is appropriate just to seek the parental consent [18]. Still, although children’s legal representatives may need to decide for them, the assent of the child should also be obtained, as far as possible [14,23]. Since emotional distress and pain are the main reasons for parents and children declining to consent, it is essential to seek consent from both of them and permit them to articulate their concerns [18]. Additionally, explaining to the parents and children who have yet not seen students, what performing a medical examination implies, is also fundamental [18].

Participants interviewed in Psychiatry were the ones who reported a lower rate of medical students clarifying their doubts, which may not be necessarily due to the students’ conduct, but rather to the distinct dynamics inherent to these consultations, where students have a more passive intervention and observational role. These participants were also the ones who felt more uncomfortable with the students. Additionally, Psychiatry participants would be more bothered by the presence of medical students if the problem was related to an intimate part of their body and were the ones claiming to have more difficulty to in revealing an intimate problem, such as sexuality, drug use or other personal matters, in their presence. Lastly, these participants would be more comfortable without the students.

As in Pediatrics, the consent for the medical students’ involvement in the Psychiatry department, for patients incompetent to make an informed decision, must be obtained from their legal representatives, while every patient’s views should be taken into account, as far as possible [23]. However, it is known that adult patients’ willingness to accept students decreases with the invasiveness of their involvement [17].

In fact, as in this study’s results, the literature shows that adult patients are much less willing to have students participate in consultations that involve worrying test results, emotional upset, sexual problems and internal examinations [17]. Actually, adult patients are more likely to refuse the students’ presence, if their complaint is of a sensitive nature, whether sexual or emotional [18], as it can happen in Psychiatry. Thus, many students may not seek consent, as they anticipate patients’ embarrassment [17]. Hence, it is essential that appropriate information is provided and consent is obtained, and that students actively assess how comfortable the patient is; if discomfort is perceived, they should have a low threshold for disengaging their participation [14].

The gender of Psychiatry patients was found to be significantly associated with their acceptance of medical students in health care, as it was the male patients who felt more uncomfortable with the students, who would be more bothered by that presence if the problem was related to an intimate part of their body and who would be more comfortable without them. Actually, it is possible that male patients feel more vulnerable exposing and expressing their sensitive issues due to the stereotype of virility that society may hold and expect from men [24]. It was also found that the age of patients from Psychiatry was significantly associated with their acceptance of the medical students’ involvement, as it was the younger patients who felt more uncomfortable with the students. In fact, a smaller age difference between patients and students might be the reason for this discomfort [25].

In addition, the education level of Psychiatry patients was found to be significantly associated with both students’ conduct and patients’ acceptance of the medical students’ involvement. Actually, those whom students least informed and least often asked for consent were higher educated patients. Moreover, it was the patients with higher education who felt more uncomfortable with the students and who would be more bothered by their presence if the problem was related to an intimate part of their body. The fact that higher educated patients can be more aware of their rights, may explain the feeling of discomfort with students, in a singularly personal setting, if a proper informed consent is not requested.

Although a considerable percentage of patients have had experiences with the medical students’ participation without their informed consent, they feel satisfied to contribute to the students’ training. As in this study’s findings, previous studies suggest that patients, usually, feel positive about participating in medical education and gain satisfaction from contributing to the students’ development [7]. Actually, altruism seems to be one of the main motivations for the patients’ participation in medical education [1]. Moreover, another common reason for patients to assist in the students’ training is the understanding that medical students need to acquire practical skills, so they practice effectively and safely when they graduate [18]. Therefore, many patients feel a responsibility to help students learn clinical skills [18], as they recognize that an effective health care system implies that medical students receive the opportunity to practice [16]. In addition, patients also enjoy being the expert in their condition [7] and learning more about their illness [18].

On the other hand, this study showed that, sometimes, there might be too many medical students in the consultation room. A proper number of students should be always sought, in order to avoid causing patients’ discomfort. Confidentiality can be another problematic aspect; many patients do not realize that agreeing to the medical students’ presence, usually, means that they will see confidential notes [15]. Thereby, according to the literature, patients should discuss with the physician which of their records may be shared and students should ask for consent to consult them, and must respect the confidentiality of medical information [14].

As more active participants, patients are increasingly exercising their partnership in care [22]. Regarding medical education, the partnership entails moving away from a position where patients are observed by the student and analyzed afterwards, to an active mode, where they take part in the discussion, during the consultation [15]. For this reason, it might be important for students to meet with patients before the consultation, making sure patients understand the primarily educational purpose of their participation [22], as well as after the consultation, in order to discuss whether students have achieved their educational goal [15]. Moreover, the medical students’ education would be improved, as they could develop an understanding of patients’ views and beliefs and begin to learn how to work in partnership with the patient, for their own later practice [15].

In fact, the literature shows that patients feel more empowered and, therefore, more ready to engage in medical teaching, if they have appropriate and understandable information, a chance to express their views and their feedback is valued [7]. An important precondition for patients’ acceptance of medical students appears also to be that they are given the opportunity to consent to or decline the students’ involvement [16]. Thereby, patients are more willing to take part in the students’ teaching, if they have their rights respected and their autonomy preserved, in particular, through the process of informed consent [4]; otherwise, they might feel aggrieved. Additionally, if patients’ concerns can be met, and their benefits increased, their willingness to participate in medical education will be optimized [15].

This study had a few limitations. Firstly, it relied on the memory of patients’ past experiences regarding the participation of medical students and how they were actually informed and asked for consent, which might always be a problem. In addition, this study was carried out in just one hospital, meaning that further multicentric studies might be needed.

The data on the absence of informed consent verified in an important percentage of participants should guide future changes. Therefore, there is a need to intervene and provide medical students with both adequate education on ethical values in clinical practice and good role models, whose behavior is supported by practices of valid and meaningful consent. Clinical teachers should set an example [11].

From the beginning of their education, medical students need guidance in building their interpersonal skills and developing a patient-centered approach to health care [2]. Thus, training in medical ethics should be initiated early in the students’ course and sustained throughout the basic and clinical years [26]; medical students must learn about the ethical dilemmas that they might face during their later practice and professional careers [7]. Thereby, the medical schools should provide a continuum of ethics education, reinforcing central concepts and essential human qualities, and, ultimately, contributing to the formation of better future doctors [2].

In fact, clinical medicine has such a practical edge that students can witness the outcome of their academic work, through their educational contact with the patient, and can experience the range of human intellect, emotion and character, embodied in the patients from whom they will learn [22]. It is unquestionable that medical students gain valuable clinical experience from their participation in health care and that they have an obligation to respect patients’ rights, throughout their care [4]. The current thinking about the patients’ involvement in medical teaching is encapsulated in the right of autonomy and principle of informed consent [18].

According to the literature, most of what students learn and internalize concerning the core values and attitudes of medicine and medical work occurs not so much in the formal curriculum, but through a more latent one. The “hidden curriculum” can often be in direct conflict with the desirable standards of ethical conduct and what is taught in the medical ethics courses [26]. In fact, many students learn to apply ethical principles, such as informed consent, in a social field, constituted by interactions between teachers, students, patients, the culture of the school and hospital [3]. Thus, the college’s members need to identify the hidden curriculum that permeates their institution and implement the best possible approaches [26].

## 5. Conclusions

Patients feel satisfied to contribute to the students’ training, although a considerable percentage of them had experiences of this without being informed or asked for consent, which represents an evident disrespect for their autonomy. Therefore, there is a need to intervene and provide an adequate education on ethical values in clinical practice to medical students, and teachers should become clear models of humanistic conduct.

It is important that the medical school examines its attitudes towards informed consent and teaches students all the necessary skills in this regard [1], as well as supplies clinical teachers with adequate training, if needed, in order to assure an optimized assistance to their students. These skills can be taught in a theoretical ethics class or, for instance, in practical sessions with mock patients, where students can be trained in requesting consent for diagnostic and therapeutic interventions, as a routine and fundamental part of patient care [7,11]. In addition, there should be specific education in the ethical conduct on teaching and learning, and teachers should always emphasize the meaning and importance of the ethical principles, maintaining excellence standards of medical education and, ultimately, of professionalism [7].

Further research could be undertaken by professional organizations and teaching institutions, in order to standardize teaching about the request for informed consent and to develop detailed guidelines that ensure the ethically acceptable practice of medical education. In fact, the Patient’s Rights in Medical Education Policy already provides a guideline for clinical teachers, that guarantees the ethico-legal practice of the interaction between students and patients, during training [4]. Moreover, medical students could also benefit from guidelines on how to approach patients, given that variables, such as patients’ demographics and culture, should always be considered [3].

Concluding, teaching hospitals are unique environments designed to carry out major education and research, while also offering high-quality health care. Both the teaching hospital and its affiliated college share the responsibility of preserving hospital schools as sanctuaries of respect for human rights and dignity [22]. Thus, the hospital holds the obligation to provide an ethical care to the patients it admits, which includes obtaining consent for the students’ participation and, concomitantly, the medical school must assume responsibility for the training and supervision of students in health care, which implies the request of informed consent from patients [4,6].

## Figures and Tables

**Table 1 healthcare-11-01270-t001:** Sociodemographic and other characteristics of the participants, per department.

	Pediatrics (*n* = 185)	Psychiatry (*n* = 185)
Participant Who Answered *n (*%*)*		
Patient	29 (16)	168 (91)
Legal Representative	156 (84)	17 (9)
Patient’s Gender *n (*%*)*		
Male	91 (49)	58 (31)
Female	94 (51)	127 (69)
Legal Representative’s Gender *n (*%*)* *		
Male	23 (15)	3 (18)
Female	133 (85)	14 (82)
Age *Mean (Standard Deviation)*		
Patient	10 (5)	45 (18)
Legal Representative	40 (7)	50 (9)
Civil Status of Respondent *n (*%*)* **		
Single	47 (25)	63 (34)
Married or Civil Union	116 (63)	86 (47)
Divorced or Widowed	22 (12)	36 (19)
Education Level of Respondent *n (*%*)* **		
1 to 4 Years	15 (8)	44 (24)
5 to 12 Years	146 (79)	85 (46)
Higher Education	24 (13)	56 (30)

* only for patients with a legal representative (*n* = 156 in Pediatrics and *n* = 17 in Psychiatry). ** *n* = 185 for each department, since these results include both patients’ and legal representatives’ answers.

**Table 2 healthcare-11-01270-t002:** Frequency (%) of answers from patients aged 16 to 18 years-old, in Pediatrics (*n* = 22).

	Yes	No	I Don’t Remember
The doctor asked me if I minded that the students were present.	11 (50)	10 (45)	1 (5)
The doctor introduced the medical students with their name and year of training.	8 (36)	13 (59)	1 (5)
I felt uncomfortable with the students’ presence.	3 (14)	19 (86)	-
Students introduced themselves as medical students and requested my consent to perform the medical examination.	9 (41)	11 (50)	2 (9)
Students explained to me the procedures they wanted to perform and clarified my doubts.	7 (32)	13 (59)	2 (9)
When medical students are present, I feel that I get more explanations about my illness.	5 (23)	17 (77)	-
If my problem was related to an intimate part of my body, I would be more bothered by the presence of medical students.	7 (32)	15 (68)	-
In case of feeling bothered by the presence of medical students, I would feel comfortable expressing my discomfort. *	5 (71)	2 (29)	-
I’m afraid to reveal an intimate problem of mine in the presence of medical students.	5 (23)	17 (77)	-
If I could choose, I would be more comfortable without the presence of medical students.	4 (18)	18 (82)	-

* only 7 answers.

**Table 3 healthcare-11-01270-t003:** Mean (standard deviation) of patients’ age for each response (e.g., felt uncomfortable with the students’ presence), in Psychiatry (*n* = 112).

	Yes	No	*p*
The doctor asked me if I minded that the students were present.	45 (15)	51 (16)	0.056
The doctor introduced the medical students with their name and year of training.	46 (16)	47 (15)	0.614
I felt uncomfortable with the students’ presence.	35 (11)	49 (16)	<0.001
Students introduced themselves as medical students and requested my consent to perform the medical examination.	49 (16)	45 (15)	0.144
Students explained to me the procedures they wanted to perform and clarified my doubts.	52 (17)	45 (15)	0.056
When medical students are present, I feel that I get more explanations about my illness.	47 (15)	46 (16)	0.813
If my problem was related to an intimate part of my body, I would be more bothered by the presence of medical students.	43 (17)	47 (15)	0.151
In case of feeling bothered by the presence of medical students’ presence, I would feel comfortable expressing my discomfort.	41 (16)	45 (17)	0.450
I’m afraid to reveal an intimate problem of mine in the presence of medical students.	43 (16)	47 (15)	0.271
If I could choose, I would be more comfortable without the presence of medical students.	42 (16)	47 (15)	0.103

**Table 4 healthcare-11-01270-t004:** Frequency (%) of affirmative answers by patients’ level of education, in Psychiatry (*n* = 112).

	4 or Less Years of School Attendance	Between 5 and 12 Years of School Attendance	Higher Education	*p*
The doctor asked me if I minded that the students were present.	14 (56)	40 (79)	27 (79)	0.124
The doctor introduced the medical students with their name and year of training.	18 (72)	33 (67)	20 (59)	0.658
I felt uncomfortable with the students’ presence.	0 (0)	10 (19)	12 (34)	0.003
Students introduced themselves as medical students and requested my consent to perform the medical examination.	13 (52)	17 (35)	4 (12)	<0.001
Students explained to me the procedures they wanted to perform and clarified my doubts.	6 (24)	12 (23)	1 (3)	0.061
When medical students are present, I feel that I get more explanations about my illness.	2 (8)	9 (18)	3 (9)	0.215
If my problem was related to an intimate part of my body, I would be more bothered by the presence of medical students.	4 (16)	16 (31)	19 (54)	0.013
In case of feeling bothered by the presence of medical students’ presence, I would feel comfortable expressing my discomfort.*	2 (50)	10 (64)	9 (47)	0.827
I’m afraid to reveal an intimate problem of mine in the presence of medical students.	2 (8)	8 (16)	12 (34)	0.056
If I could choose, I would be more comfortable without the presence of medical students.	3 (12)	16 (31)	14 (40)	0.126

Here the (percentages) represent the percentage of answers “yes” for each sentence, separated by patients’ level of education. *****
*n* = 39.

**Table 5 healthcare-11-01270-t005:** Frequency (%) of affirmative answers by department (*n* = 112). “I don’t remember” answers were excluded.

	Pediatrics	Psychiatry	*p*
The doctor asked me if I minded that the students were present	68 (54)	88 (75)	<0.001
The doctor asked the person I represent if he or she minded that the students were present.	23 (22)	4 (67)	0.032
The doctor introduced the medical students with their name and year of training.	50 (41)	76 (65)	<0.001
I felt uncomfortable with the students’ presence.	5 (4)	22 (18)	<0.001
I felt the person I represent uncomfortable with the students’ presence.	6 (6)	3 (38)	0.015
Students introduced themselves as medical students and requested my consent to perform the medical examination.	44 (36)	37 (32)	0.528
Students introduced themselves as medical students and requested the person I represent’s consent to perform the medical examination.	24 (24)	3 (50)	0.164
Students explained to me the procedures they wanted to perform and clarified my doubts.	45 (35)	19 (16)	<0.001
Students explained to the person I represent the procedures they wanted to perform and clarified his or her doubts.	35 (33)	0 (0)	0.175
When medical students are present, I feel that I get more explanations about my illness/illness of the person I represent.	33 (25)	16 (13)	0.016
If my problem/problem of the person I represent was related to an intimate part of my body, I would be more bothered by the presence of medical students.	26 (20)	41 (34)	0.012
In case of feeling bothered by the presence of medical students’ presence, I would feel comfortable expressing my discomfort.	21 (81)	23 (56)	0.038
I’m afraid to reveal an intimate problem of mine/intimate problem of the person I represent in the presence of medical students.	11 (8)	22 (18)	0.021
If I could choose, I would be more comfortable without the presence of medical students.	9 (7)	37 (31)	<0.001

Here the (percentages) represent the percentage of answers “yes” for each sentence, separated by department.

## Data Availability

The datasets used and analyzed during the current study are available from the corresponding author on a reasonable request.

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
