# Peer review of "Consent for Teaching—The Experience of Pediatrics and Psychiatry"

_healthcare, 2023, doi:10.3390/healthcare11091270_

Round 1

Reviewer 1 Report

This is an interesting, important and relevant paper dealing with informed consent for teaching in pediatrics and Psychiatry. The paper extensively elaborates on the topic.

There are three main issues that need to be considered:

1. The question is, if the length of the paper can be reduced. 2 pages of introduction is very long. Part of the introduction is again repeated in the discussion. Taking this into consideration the paper can be optimized substantially.

2. It is important to know, which information was given to students before they had contact to patients. Were they informed, to ask for consent or not. Was there an introduction given by teachers how to address patients or guardians? Did students trained the process of informed consent with actors/manekins beforehand? And in addition, did the teachers demonstrate the process of informed consent with patients or guardians in front of students? This may also be a solution to get always informed consent.

3. Was there a difference in the percentage of getting informed consent between invasive procedures done by students (like putting an i.v. or taking blood) and non-invasive procedure as inspection, palpation etc.?

Other issues:Tables are difficult to read. In part of the tables the description of a line is not in the same heigh as the numbers for each category. In some tables one cannot understand the percentages given for each column and line, if the absolute number is only given for all columns but not for each column seperately. This should be changed.

Author Response

Reviewer 1

  1. The question is, if the length of the paper can be reduced. 2 pages of introduction is very long. Part of the introduction is again repeated in the discussion. Taking this into consideration the paper can be optimized substantially.

Response/Actions: Thank you for pointing this out. We understand your concern and we were able to reduce the introduction. Here’s the information we eliminated:

  • Lines 30 and 31: “its contribution is fundamental to the complete education of medical students, in both clinical and ethical components.;
  • Lines 37 and 38: “contact with real patients and”;
  • Line 53 to 55: “These are fundamental ethical principles for the medical practice and should be the very pillars of Ethics delivered to medical students as foundations to their future clinical practice.”;
  • Line 68 to 71: “Thereby, informed consent must be free from coercion, manipulation or persuasion and patients should know the nature and purpose of the examination, the status and qualifications of care providers and the planned extent of their intervention, before making a decision [7].”;
  • Line 80 to 82: “Thus, both students and clinical teachers should be aware that patients may refuse to participate in the medical students' learning and education, and decline students’ assistance in their health care provision. [16]”;
  • Line 85: “and power”;
  • Line 87 to 89: “Moreover, patient’s legal representatives can too be included in the development of this process, in order to ensure high standards of ethical practice in medical education [1].”

  1. It is important to know, which information was given to students before they had contact to patients. Were they informed, to ask for consent or not. Was there an introduction given by teachers how to address patients or guardians? Did students trained the process of informed consent with actors/mannequins beforehand? And in addition, did the teachers demonstrate the process of informed consent with patients or guardians in front of students? This may also be a solution to get always informed consent.

Response/Actions: Thank you for your concern. We understand and agree that´s an important and interesting matter, but it’s not part of the scope of this study. We only focused on the patients’ point of view; therefore, we did not questioned/interviewed any students, and do not have the information to answer such questions. That would surely be an excellent subject to be considered in a new study, which could focus on students’ perspectives and education.

  1. Was there a difference in the percentage of getting informed consent between invasive procedures done by students (like putting an i.v. or taking blood) and non-invasive procedure as inspection, palpation etc.?

Response/Actions: Once again, that was not the purpose of this particularly study, as invasive procedures performed by medical students was not the main concern in the medical specialties considered, being that more probably to happen in both surgical and adult specialties. That would actually be another great matter to study and compare the obtainment of informed consent between some of the main medical and surgical specialties.

Other issues: Tables are difficult to read. In part of the tables the description of a line is not in the same heigh as the numbers for each category. In some tables one cannot understand the percentages given for each column and line, if the absolute number is only given for all columns but not for each column separately. This should be changed. 

Response/Actions: We appreciate this note. We were able to change the tables, by altering some aspects in the content and configuration, in order to improve their reading and understanding.

Reviewer 2 Report

In my opinion, this is an article of interest for bioethical reflection on clinical practice. It has a good methodology, formal development, good knowledge of the bibliography on the subject studied, adequately analyzes the results, and makes weighted ethical assessments.

There is something that, throughout the reading and analysis of the study, I would have liked to know: a simple comparison with the opinion of autonomous and capable adult patients, for example, Internal Medicine (for choosing a more general department). It would be a question of knowing if, on the one hand, the information given by doctors or students is more or less if people with potentially compromised autonomy are treated; and, on the other hand, if pediatric or psychiatric patients consent to a greater or lesser extent to the participation of medical students.

In conclusion, it is a good article as it is, my contribution only offers new data for reflection, it does not diminish the quality of the research.

Author Response

Reviewer 2

In my opinion, this is an article of interest for bioethical reflection on clinical practice. It has a good methodology, formal development, good knowledge of the bibliography on the subject studied, adequately analyzes the results, and makes weighted ethical assessments.

There is something that, throughout the reading and analysis of the study, I would have liked to know: a simple comparison with the opinion of autonomous and capable adult patients, for example, Internal Medicine (for choosing a more general department). It would be a question of knowing if, on the one hand, the information given by doctors or students is more or less if people with potentially compromised autonomy are treated; and, on the other hand, if pediatric or psychiatric patients consent to a greater or lesser extent to the participation of medical students.

In conclusion, it is a good article as it is, my contribution only offers new data for reflection, it does not diminish the quality of the research.

Response/Actions:We really appreciate your comment. In fact, we conducted a study previous to this one in the same institution, concerning patients' perspectives about medical students' involvement in their medical care in the specialties of General Surgery, Obstetrics/Gynecology and Infectious Diseases. Therefore, our goal is to continue and maybe study a few more medical and surgical specialties regarding this subject, and we consider it would be really interesting and relevant to, in the future, compare the data between all of the departments and analyze those findings.